# The Influence of Social Support and Care Burden on Depression among Caregivers of Patients with Severe Mental Illness in Rural Areas of Sichuan, China

**DOI:** 10.3390/ijerph16111961

**Published:** 2019-06-02

**Authors:** Xiaxia Sun, Jingjing Ge, Hongdao Meng, Zhiguo Chen, Danping Liu

**Affiliations:** 1Department of Health Related Social and Behavioral Science, West China School of Public Health, Sichuan University, Chengdu 610041, China; sunxiaxia@163.com (X.S.); 18280085667@163.com (J.G.); 2School of Aging Studies, College of Behavioral & Community Sciences, University of South Florida, Tampa, FL 33620, USA; meng@usf.edu; 3Department of Biostatistics, University of Florida, Gainesville, FL 32611, USA; bruchen@ufl.edu

**Keywords:** primary caregivers, care burden, social support, depression, severe mental illness, rural areas

## Abstract

Depression is one of the most common psychological consequences of caregiving. Caring for patients with severe mental illness (SMI) adds significant challenges to family caregivers’ mental health. The purpose of this study was to describe the prevalence of depression among caregivers of SMI patients in rural areas of Sichuan province of China, to examine the influence of social support and care burden on depression, and to explore the intermediary effect of care burden between social support and depression among caregivers of SMI patients. Data were collected from 256 primary caregivers of SMI patients in rural Sichuan Province in China. We used structural equation modeling (SEM) to test the hypothesized relationship among the variables. We found that a total of 53.5% of caregivers had depression. Both care burden (β = 0.599, 95%CI: 0.392–0.776) and social support (β = −0.307, 95%CI: (−0.494)–(−0.115)) were directly related to depression, while social support had a direct association with care burden (β = −0.506, 95%CI: (−0.672)–(−0.341)). Care burden mediated the relationship between social support and depression. For the socio-demographic variables, gender, education level and per capita annual income of household had significant correlations with depression (*p* < 0.05). The results strongly demonstrated that social support and care burden were predictors of depression, especially social support. Policymakers should fully recognize the role of primary family caregivers in caring for SMI patients and promote interventions to decrease care burden and reduce caregivers’ depression by improving social support and network. More attention should be given to female caregivers and caregivers with lower education and lower household income levels.

## 1. Introduction

The term severe mental illness (SMI) refers to a variety of mental health disorders, but typically illnesses classified as SMI include schizophrenia and schizophrenia-related disorders, bipolar disorder, major recurrent depressive disorder and personality disorders [1]. In China, there were 5.8 million registered patients with SMI until the end of 2017. Over 90% of those individuals were in a stable or basically stable condition and were recommended to receive long-term treatment and rehabilitation in the community instead of in medical institutions [2]. In addition, though China has not experienced the de-institutionalization that aimed to shift the care responsibility of psychiatric care from the formal health system to family caregivers [3], the Chinese traditional Confucian culture strongly advocates for family harmony and integrity, especially focusing on caring for the impaired family members. As a result, in China, family members of SMI patients take on multiple roles in caring for ill relatives.

Family caregivers of SMI patients often experience tremendous financial and social distress, emotional burden and mental distress [4]. These challenges may increase caregivers’ vulnerability to serious mental health problems, and caregivers may experience serious mental health problems [5,6]. Depression has been cited as one of the main psychological consequences of caregiving [7,8]. Studies have found that the depression rate among caregivers of SMI patients was more than two times higher than that of the general population [9,10].

As a common mental disorder, depression can be chronic or recurrent, thus substantially impairing an individual’s ability to function in their daily life [11]. Mental disorders have also been identified as significant risk factors for both suicidal behavior and suicidal ideation, especially among depressed patients [12]. Pearlin et al. proposed and developed the stress process model of depression [13] in 1981. Based on that, extensive research had investigated the influencing factors of depression not only in the general population [14,15] but also in the caregiver population. Factors associated with caregivers’ depression mainly include behavioral problems of care recipients [16], caregivers’ socio-demographic factors [17], social support [18] and care burden [19,20]. Different combinations of these factors were often incorporated in studies to explore the combined effect on depression [21,22].

Social support is defined as the interpersonal resources that individuals accessed and mobilized when they attempt to deal with the daily stresses and strains of life [23]. Previous studies demonstrated that increased risk of depression status was associated with the lack of social support among different groups (like elderly people [24] and patients with Parkinson’s disease [25]). Hobbs found that social support was a significant predictor of depression among caregivers of SMI patients [26]. Saunders showed that insufficient social support was the most significant predictor of depression among caregivers of SMI patients, care burden followed [21]. Social support also could reduce the care burden [18,27].

Care burden usually refers to the physical, psychological, financial, and social discomfort that are experienced by the principal caregiver of a disabled family member [28]. Care burden could be objective, measured by the impact of caring for patients on family resources. It could also be subjective, that is, the mental health and emotional impacts that were felt while caregivers care for ill relatives [29,30]. Family caregivers of patients with SMI experienced a high-level of care burden [31,32]. Care burden has been frequently considered as a strong predictor of depression in family caregivers who care for individuals with SMI [21,33] as well as a mediator of the relationship between other factors and depression. For example, care burden fully mediated the effect of social support on levels of depressive symptoms among caregivers of lung cancer in a study by Kim Y et al. [22].

The mental health of caregivers is critical to the person being cared for [34]. Given that depression is one of the main causes of disability and disease burden worldwide [35], higher levels of depression will influence family caregivers’ ability to care for SMI patients. Analyzing how social support and care burden affect depression of caregivers of SMI patients is important for promoting the health of caregivers and even SMI patients. However, very little is known about the interrelationships and potential mechanisms of social support, care burden and depression among caregivers of SMI patients.

Based on the previous research, we examined the influence of social support and care burden on the depression of caregivers of SMI patients in rural areas of Sichuan province of China. We hypothesized a single mediator model shown in Figure 1. Specifically, social support would be negatively associated with both care burden (hypothesis 1) and depression (hypothesis 2). We also hypothesized that the care burden would be positively associated with depression (hypothesis 3). In addition, we suggested that the relationship between social support and depression would be mediated by the care burden (hypothesis 4). This study is the first to explore the influence of social support and care burden on depression among caregivers of SMI patients in China. Findings from this study may have important implications for strategies to decrease care burden and depression and promote the health of family caregivers of SMI patients.

## 2. Materials and Methods

### 2.1. Settings and Participants

This cross-sectional research was conducted among caregivers of SMI patients in Sichuan province, Southwest China from December, 2017 to May, 2018. The primary family caregiver of each selected SMI patient was selected as the target population. The primary family caregiver was defined as the family member who was more than 18 years, spent the most time with the SMI patient and provided the most care to the SMI patient during the prior year.

A multi-stage stratified random sampling was used to acquire the sample. In the first stage, we randomly chose a city in Sichuan province. In the second stage, we randomly selected a rural district in the city. In the third stage, ten townships were randomly selected from the rural district. In the fourth stage, we randomly selected 30 registered SMI patients from each township. A total of 300 primary caregivers were eligible to participate in the study and 256 caregivers completed the questionnaire (for an effective response rate of 85.3%).

### 2.2. Ethical Consideration

The study protocol was approved by the Institutional Review Board of Sichuan University (Project identification code: H171260). Informed consent was obtained from each primary caregiver following a detailed explanation about the purpose of the study.

### 2.3. Measures

Caregivers’ socio-demographic characteristics, social support, care burden and depression information were collected from questionnaires.

#### 2.3.1. Socio-Demographic Characteristics

Socio-demographic characteristics included age, gender, marital status, education level, employment status, individual annual income, per capita annual income of the household and medical insurance status.

#### 2.3.2. Social Support

Social support was measured by the Chinese version of the social support rating scale (SSRS), which was developed by Xiao S.Y. et al. [36]. The SRSS has been used in Chinese caregiver populations and demonstrated to have high scale reliability and consistency [20]. The SSRS is a self-reported scale including ten items and three domains: subjective support, objective support and use of social support. The total score of social support is the sum of scores of all items (ranges from 12 to 66) with a higher score reflecting higher social support. The total score has been divided into three levels: low (12–22), moderate (23–44) and high (45–66). In the current study, Cronbach’s alpha of the scale was 0.825.

#### 2.3.3. Care Burden

Care burden was assessed by the Zarit Caregiver Burden Interview (ZBI) [23]. It is a 22-item survey classified into five dimensions including negative emotion, interpersonal relationship, time demand, patient’s dependence, self-accusation and guilt [37]. Items are rated on a 5-point Likert scale from 0 (never) to 4 (nearly always). The total score was calculated by adding the response score for each item and ranged from 0 to 88 (less than 21 = little or no burden; 21–40 = mild burden; 41–60 = moderate burden; 61–88 = severe burden) with higher scores indicating higher care burden. The ZBI has been most widely used and validated in caregivers in China [38]. In the current study, Cronbach’s alpha of the scale was 0.933.

#### 2.3.4. Depression

Depression was measured by the Chinese version of the ten-item Center for Epidemiologic Studies Depression Scale (CES-D 10) [39,40] and the Cronbach’s alpha coefficient was 0.87 [41]. The CES-D 10 includes depressive affect and somatic symptoms, and positive affect [42]. Four-point Likert scale ranging from 0 (experienced rarely or none of the time) to 3 (experienced most or all of the time) was utilized to evaluate all the 10 items, higher scores indicated greater depressive symptom. Total score ranges from 0 to 30, and a score of 10 or higher indicates significant depressive symptoms [43]. In the current study, Cronbach’s alpha of the scale was 0.786.

### 2.4. Statistics Analysis

Data were entered using the Epidata 3.1 database and were analyzed using the SPSS version 23.0 (SPSS Inc., Chicago, IL, USA) and Analysis of Moment Structures (AMOS) version 22.0 (IBM, New York, NY, USA).

We first used descriptive statistics to examine the socio-demographic characteristics of caregivers of SMI patients. Second, we undertook a descriptive analysis of caregivers’ social support, care burden and depression, using means and standard deviations (SD). Third, Pearson correlation coefficient was used to analyze the correlation of social support, care burden and depression. A linear regression model was also used to analyze the influence of socio-demographic factors, in addition to social support and care burden on depression status. Fourth, a structural equation model (SEM) was employed to further test the hypothesized relationships among social support, care burden and depression of caregivers of SMI patients.

The SEM used bootstrap maximum likelihood estimation and the results, with a *p*-value of < 0.05, were considered statistically significant. The fit between the current data and the hypothesized model was assessed through several indicators, adjusted goodness of fit index (AGFI), a goodness of fit index (GFI), the comparative fit index (CFI), normed fit index (NFI), incremental (IFI), and Tucker-Lewis index (TLI) of 0.90 or above. A root mean squared error of approximation (RMSEA) less than or equal to 0.08 [44], indicated an acceptable model fit.

## 3. Results

### 3.1. Socio-Demographic Characteristics of Primary Caregivers

Socio-demographic characteristics of the 256 primary caregivers are shown in Table 1. Over half (54.3%) of the caregivers were female with a mean age of 58.0 years (SD = 13.1). Most were married (81.6%), educated elementary school or less (61.3%) and had medical insurance (93.7%). The majority of the caregivers were currently employed (48.8%), had an individual annual income less than $750 (45.3%) and had a per capita annual income of household less than $750 (45.3%).

### 3.2. Descriptive Analysis of Study Variable

Table 2 shows scores of the 256 caregivers’ social support, care burden and depression. The mean score of care burden was 42.3 ± 15.9, 19 (7.4%) of caregivers had little or no burden, 103 (40.2%), 101 (39.5%) and 33 (12.9%) of caregivers had a mild, moderate and severe burden, respectively. The mean score of social support was 32.1 ± 7.7, 24 (9.4%) of caregivers experienced low social support, 217 (84.8%) and 15 (5.8%) of caregivers experienced moderate and high social support, respectively. The mean score of depression of the caregivers was 10.0 ± 5.3 and 137 (53.5%) of caregivers had depression.

### 3.3. Correlations of Study Variable

The Pearson’s correlations between care burden, social support and depression are presented in Table 3. Social support is negatively correlated with care burden and depression. Care burden is positively correlated with depression.

### 3.4. Linear Regression Analysis of Study Variable

We used depression as the dependent variable and socio-demographic variables, social support and care burden as independent variables in an enter linear regression model. Table 4 presented the statistically significant variables in the analysis (*p* < 0.05). The results showed that three socio-demographic factors in addition to social support and care burden were significantly correlated with depression: gender, educational level and per capita annual income of the household. Female caregivers were more likely to be depressed (β = 0.175, *p* = 0.002). Caregivers who were educated at middle school (β = −0.193, *p* = 0.014) and high school and above (β = −0.148, *p* = 0.048) were less likely to be depressed when compared with those educated less than elementary school. Caregivers who had a per capita annual income of the household of $1500 and above were less likely to be depressed when compared with caregivers who earned less than $750 (β = −0.231, *p* = 0.009).

### 3.5. Test of Study Model

We fitted the data and the theoretical model through the generalized least squares and modified the theoretical model according to model fit indices. With the addition of socio-demographics as covariates, the arrow direction among the core variables in the SEM remained unchanged and the corresponding coefficients did not change significantly. Thus, the socio-demographics were not confounding factors. Figure 2 shows the final model where all paths were statistically significant and the model had an adequate fit: GFI = 0.974, AGFI = 0.943, NFI = 0.966, IFI = 0.991, TLI = 0.984, RMSEA = 0.037.

Bootstrap with 2000 replications using maximum likelihood estimation was employed for each path. The estimates for direct, indirect and total effects with bias-corrected 95% CI are shown in Table 5. The results showed that social support had a significant negative correlation with care burden (β = −0.506, 95%CI: (−0.672)–(−0.341)) and depression (β = −0.307, 95%CI: (−0.494)–(−0.115)). Care burden had a significant positive correlation with depression (β = 0.599, 95%CI: 0.392–0.776).

Regarding the path between social support and depression, the total effect, indirect effect and direct effect were all statistically significant, which means there exists a mediating effect of care burden in the relation between social support and depression. Based on the above, social support was both directly and indirectly associated with depression by the mediating role of care burden. Therefore, the final results supported all of the hypotheses.

## 4. Discussion

The study investigated the depression status of caregivers of SMI patients in rural areas of Sichuan province of China. Our findings indicated a worth noticing result that the prevalence of depression among caregivers of SMI patients (53.5%) is higher than that of caregivers of SMI patients reported from Sri Lanka (37.5%) [45] and California USA (40%) [10], and also greater than that of caregivers of SMI patients in a rural central China study (45.4%) [43]. One reason for the differences may be that mental health service in the relatively backward southwest region is poorer than those in the central region of the country [46]. This may prevent SMI patients from getting efficient treatment timely [47], which adds care burden to caregivers [48] and increases depression of caregivers. Another possible explanation may be that SMI patients and their families generally encountered a substantial amount of stigma and discrimination from the public in China. Another reason may be that people avoid having too much contact with mentally ill individuals and their families [49], which may increase caregivers’ depression.

The results revealed that caregivers of SMI patients reported a moderate burden with a mean score of 42.3 ± 15.9 which is lower than that of caregivers of schizophrenia patients in Hamadan (51.7 ± 18.2) (*p* < 0.001) [50]. In this study, only 12.9% of caregivers reported severe care burden. It may due to the strong cultural norms. These norms that make it the Chinese responsibility to take care of an ill family member, may make them reluctant to admit experiencing care burden and may result in feelings of guilt associated with thoughts of care burden [51]. The concept of filial obligation may result in caregivers underreporting the burden they feel conscious. The model supported that caregivers of SMI patients with more care burden were more likely to experience depression symptoms which were consistent with previous studies. Hsiao et al. and Song et al. have demonstrated a positive correlation between care burden and depression in caregivers of SMI patients [29,52]. Therefore, to decrease depression among caregivers of SMI patients, efforts should be strengthened to decrease caregivers’ care burden.

The mean score of social support among caregivers of SMI patients was 32.1 ± 7.7 and most (84.8%) of caregivers experienced moderate social support from families and friends in this study. This is consistent with a previous study conducted on caregivers of patients with mental disorders in China [51]. The model revealed that caregivers of SMI patients with higher social support were less likely to experience depression symptoms which is consistent with previous studies. A one-year community-based prospective cohort study in Japan showed a significant increase in the risk of depression status associated with the lack of social support among the elderly [53]. One possible explanation may be that caregivers who received insufficient social support could not buffer caring stress effectively and the incidence of various kinds of related psychic disturbances, including depression, would be increased [54]. This suggests that providing sufficient social support represents a useful strategy for reducing depression of caregivers of SMI patients.

The most significant finding of this study was that the relationship between social support and depression was mediated by care burden. The model showed that social support was a significant and direct predicting factor for care burden among caregivers of SMI patients which is consistent with previous studies [55,56]. This can be explained by the buffer model, in which the distress and burden in caring for a relative with mental illness appear to be buffered by the satisfaction of social support when individuals receive sufficient social support [57,58]. Based on the above, social support not only influences depression directly but also exerts an influence on depression indirectly through the mediating role of care burden. This reveals that social support is essential to decrease depression and that interventions to reduce depression in caregivers should be inspired by this mediating path. One possible approach is to enhance caregivers’ ability to acquire and use social support from family and friends. In addition to informal support from families and friends, formal mental health services are also of great importance for SMI patients and their caregivers. However, mental health services resources in China are very deficient, and the present utilization is even more inadequate, especially in rural areas [59]. Therefore, a formal supportive network should be primarily considered.

This study also identified that female caregivers were more likely to be depressed which was consistent with previous research. Derajew et al. indicated that the prevalence of depression of female caregivers of SMI patients was higher than that in male caregivers [60]. This suggests that special attention should be paid to female caregivers. Findings from the study also revealed that two other socio-demographic characteristics of caregivers were associated with depression: education and income level, the two most common socioeconomic status (SES) factors. This supports the finding of a previous study in which SES is negatively associated with depression [61]. After controlling for other demographic characteristics, caregivers with higher income and education level were less likely to be depressed which was consistent with studies by Cummings et al. [17] and Magana et al. [10]. This may be because lower SES means fewer resources available for caregivers to reduce care burden. Thus, practical ways to assist families with lower SES may help relieve depression of caregivers of SMI patients.

Study limitations should be taken into account. First, despite the SEM being used to determine the relationship between the variables, the cross-sectional design imposes a significant limitation to drawing any definitive conclusions. In addition, while this study is among the first to examine the influence of caregivers’ social support and care burden on depression in one model in rural areas of Sichuan of China, the findings may not be generalizable to other regions in China. Finally, other factors such as loneliness and self-reported health status have not been taken into consideration, which may also impact the depression status of caregivers. Future studies should explore more factors that may influence depression.

## 5. Conclusions

The study shows that the rate of depression among primary caregivers of SMI patients is high in rural areas of Sichuan province of China. Mental health services should be provided to prevent mental illness for this special group. The results showed that insufficient social support and more care burden are predictors of higher depression of caregivers of SMI patients. In addition, social support exerts a significant influence on depression indirectly through the mediating role of care burden. The influence of social support on depression can be heightened further through lower care burden, so social support was essential to decrease depression. For the socio-demographic variables, female, lower levels of caregivers’ education and per capita annual income of household were predictive of higher levels of caregivers’ depression symptoms. The results imply that this special group should be noticed by policymakers. To decrease care burden and reduce depression in caregivers of SMI patients, measures that could act effectively on social support should be taken into consideration. At the same time, constructing a formal supportive network in addition to social support from families and friends should be considered. Greater attention to the female caregivers and caregivers who had lower education and lower household income levels is needed.

## Figures and Tables

**Figure 1 ijerph-16-01961-f001:**
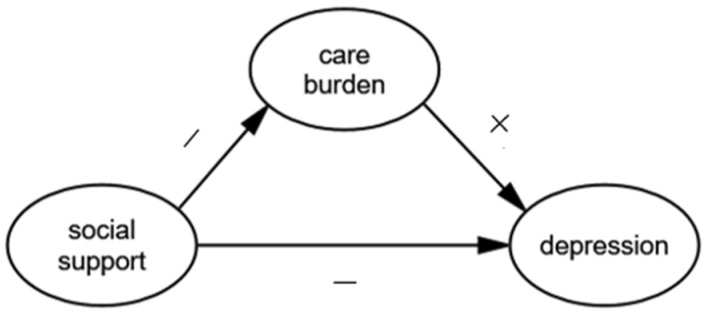
The theoretical model and hypotheses.

**Figure 2 ijerph-16-01961-f002:**
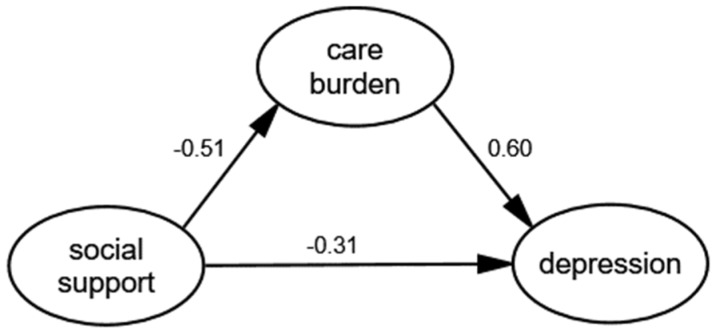
The final model and standardized model paths.

**Table 1 ijerph-16-01961-t001:** Socio-demographic characteristics of primary caregivers (*n* = 256).

Characteristics	*N* (%)
Age, mean ± SD	58.0 ± 13.1
Gender	
Male	117 (45.7)
Female	139 (54.3)
Marital status	
Single	7 (2.7)
Married	209 (81.6)
Divorced or widowed	40 (15.6)
Education level	
Less than Elementary school	51 (19.9)
Elementary school	106 (41.4)
Middle school	69 (27.0)
High school and above	30 (11.7)
Employment status	
Employed	125 (48.8)
Unemployed	86 (33.6)
Retired	45 (17.6)
Individual annual income, $	
<750	116 (45.3)
750–1500	34 (13.3)
≥1500	106 (41.4)
Per capita annual income of the household, $	
<750	116 (45.3)
750–1500	55 (21.5)
≥1500	85 (33.2)
Medical insurance	
Yes	240 (93.7)
No	16 (6.3)

**Table 2 ijerph-16-01961-t002:** Description of social support, care burden and depression scores (*n* = 256).

Contents	Range	Mean (SD)
**Care burden**	0–88	42.3 ± 15.9
Negative emotion	0–40	18.3 ± 7.6
Interpersonal relationship	0–16	7.3 ± 4.0
Time demand	0–12	5.7 ± 2.8
Patient’s dependence	0–8	4.5 ± 2.0
Self-accusation and guilt	0–8	3.9 ± 1.8
**Social support**	12–66	32.1 ± 7.7
Subjective support	8–32	18.9 ± 5.2
Objective support	1–22	7.7 ± 2.3
Use of social support	3–12	5.6 ± 2.2
**Depression**	0–30	10.0 ± 5.3
Positive affect	0–6	2.6 ± 1.8
Depressive affect and somatic symptoms	0–24	7.4 ± 4.4

**Table 3 ijerph-16-01961-t003:** Correlation coefficients among study variables.

Variables	(1)	(2)	(3)
(1) Care burden			
(2) Social support	−0.239 **		
(3) Depression	0.463 **	−0.298 **	

** *p* < 0.01.

**Table 4 ijerph-16-01961-t004:** Linear regression of factors associated with the depression.

Factors	Unstandardized Coefficients		Standardized Coefficients	*t*	*p*-Value	95%CI for β
	β	SE	β			
Constant	12.173	3.359	-	3.624	<0.001	(5.555, 18.790)
Female (ref: Male)	1.875	0.608	0.175	3.085	0.002	(0.678, 3.073)
Education level (ref: Less than elementary school)						
Elementary school	−1.295	0.844	−0.120	−1.534	0.126	(−2.959, 0.369)
Middle school	−2.311	0.937	−0.193	−2.465	0.014	(−4.157, −0.464)
High school and above	−2.451	1.231	−0.148	−1.992	0.048	(−4.876, −0.027)
Per capita annual income of household (ref: <750, $)						
750–1500	−1.932	0.993	−0.149	−1.945	0.053	(−3.889, 0.025)
≥1500	−2.615	0.999	−0.231	−2.616	0.009	(−4.584, −0.646)
Social support scores	−0.126	0.041	−0.181	−3.097	0.002	(−0.207, −0.046)
Care burden scores	0.113	0.020	0.338	5.617	<0.001	(0.074, 0.153)

Notes: R^2^ = 0.34, F = 7.51, *p* < 0.001.

**Table 5 ijerph-16-01961-t005:** The path coefficients between structural variables.

Pathways	Estimate	95%CI
**Total effects**		
Care burden←Social support	−0.506	(−0.672, −0.341)
Depression←Social support	−0.610	(−0.763, −0.450)
Depression←Care burden	0.599	(0.392, 0.776)
**Direct effects**		
Care burden←Social support	−0.506	(−0.672, −0.341)
Depression←Social support	−0.307	(−0.494, −0.115)
Depression←Care burden	0.599	(0.392, 0.776)
**Indirect effects**		
Depression←Social support	−0.303	(−0.443, −0.198)

Abbreviation: CI, confidence interval.

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
