# Peer review of "The Influence of Social Support and Care Burden on Depression among Caregivers of Patients with Severe Mental Illness in Rural Areas of Sichuan, China"

_ijerph, 2019, doi:10.3390/ijerph16111961_

Round 1
Reviewer 1 Report
I have highlighted areas where words are missing, grammar or syntax is incorrect, and the meaning you intended is lost. There are multiple places that would be confusing for English as a first language readers.You also have several run-on sentences. There is one incomplete sentence (see line #250-252).
In research, we do not say that any model or hypothesis is "proved" or proven, we say that it is supported by the statistics.
You have an interesting study, with important information. However, it really needs a careful rewrite.

Author Response
Point 1: I have highlighted areas where words are missing, grammar or syntax is incorrect, and the meaning you intended is lost. There are multiple places that would be confusing for English as a first language readers. You also have several run-on sentences. There is one incomplete sentence (see line #250-252). Response 1: Thank you for your correction. As we proofread the manuscript in detail, we checked and corrected all of the words missing, grammar or syntax errors, lost meaning, run-on sentences and incomplete sentences in our manuscript accordingly as possible as we can (see details in the revised version of manuscript). Thus, the incomplete sentence in line #250-252 has also been corrected (page 8, line 233-236). Point 2: In research, we do not say that any model or hypothesis is "proved" or proven, we say that it is supported by the statistics. Response 2: Thank you for your correction. we have replaced sentences like “the model proved……” with “ the model supported that……” (page 8, line 247). Point 3: You have an interesting study, with important information. However, it really needs a careful rewrite. Response 3: Thank you for your kind advice. We have rewrote the manuscript carefully according to your suggestions.
Reviewer 2 Report
Sun et al.: The Influence of Social Support and Care Burden on Depression among Caregivers of Patients with Severe Mental Illness in Rural Areas of Sichuan, China
This is a cross-sectional study aiming to describe the prevalence of depressions among caregivers of patients with severe mental illness (SMI) in rural areas of Sichuan province of China and to examine the influence of social support and care burden on depression, and to explore the intermediary effect of care burden between social support and depression among caregivers of SMI patients. The study is well designed. Sample selection was correct.
Detailed comments:
Results, socio-demographic characteristics section: repeating the data in the text that is shown in Table 1 is not necessary.
Results, socio-demographic characteristics section: concerning income, please give the amount in dollars to inform non-Chinese readers.
Results, linear regression analysis of study variable section: Table 4 with the detailed results can be deleted. Publising significant results in the text is enough.
Conclusion, 2nd sentence: I have doubts wether mental health services can prevent mental illnesses of the caregivers. Mental health professionals can screen caregivers for early diagnosis and can adequatelly treat mental disorders but cannot prevent them.
Author Response
Point 1: Results, socio-demographic characteristics section: repeating the data in the text that is shown in Table 1 is not necessary.
Response 1: Thank you for your kind advice. We rewrote the “Results, socio-demographic characteristics section” and described the data shown in Table 1 more succinctly (page 4, line 171-175).
Point 2: Results, socio-demographic characteristics section: concerning income, please give the amount in dollars to inform non-Chinese readers.
Response 2: Thank you for your kind advice. We have given the amount of individual annual income and per capita annual income of household in dollars to inform non-Chinese readers (page4, line 174-175; page 5, Table 1; page 6, line 201-203; page6, Table4).
Point 3: Results, linear regression analysis of study variable section: Table 4 with the detailed results can be deleted. Publising significant results in the text is enough.
Response 3: Thank you for your kind advice. We have deleted the detailed results of linear regression analysis shown in Table 4 and just showed the statistically significant variables in the text (page 6, line 194-195; page 6,Table 4).
Point 4: Conclusion, 2nd sentence: I have doubts wether mental health services can prevent mental illnesses of the caregivers. Mental health professionals can screen caregivers for early diagnosis and can adequatelly treat mental disorders but cannot prevent them.
Response 4: Thank you for your question. In China, primary medical institutions, like community health centers and stations in urban areas and township hospitals and village clinics in rural areas, are required to provide primary mental health services for the public. These services including education and promotion of mental health knowledge, psychological consultation and guidance, mental health survey and documentation and so on[1], which can enhance residents’ self-adjustment ability and prevent the occurrence of mental disorders. Therefore, providing mental health service can also prevent mental disorders like depression in caregivers of SMI patients.
Reference
1. General Office of the State Council. National mental health work plan (2015-2020).http://www.nhc.gov.cn/jkj/s5888/201506/1e7c77dcfeb4440892b7dfd19fa82bdd.shtml (May 6th, 2019)

Reviewer 3 Report
This is, in summary, an interesting manuscript aimed to describe the prevalence of depression among 256 primary caregivers of SMI patients in rural areas of Sichuan province of China and examine the influence of social support and care burden on depression as well as explore the intermediary effect of care burden between social support and depression in the same population. The authors reported that 53.5% of the caregivers had depression. Care burden had a positive effect on depression, social support had a negative effect on care burden, social support had both direct negative effect and indirect negative effect on depression through the mediating role of care burden as well. Additionally, per capita annual income of household had significant correlations with depression. The authors added that more social support and lower care burden would be useful to reduce depression of caregivers of SMI patients, especially the effect of social support on decreasing depression may be heightened further through lower care burden, with social support that was essential to decrease depression.
The authors may find as follows my main comments/suggestions.
First, when the authors, within the Introduction section, reported that as a common mental disorder, depression may be long-lasting or recurrent, thus substantially impairing the individual’s ability to function in their daily life, they could also mention that depression is commonly associated with suicidal behavior and medical illnesses related to a further ehnanced suicide risk. In particular, medical disorders have been identified as significant risk factor for both suicidal behavior and suicidal ideation, especially among depressed patients. Specifically, the existence of previous mood disorder, prior and current history of medical disorders, and cognitive impairment were reported to be the most important risk factors for suicide. Thus, in order to briefly address this issue (although i understand that the link between medical disorders, depression, and suicidal behavior is not the main topic of this paper), i suggest to cite and rapidly discuss the systematic review paper published on Drugs Aging in 2015 (PMID: 25491561).
Moreover, the most relevant psychometric instruments used in the present study (e.g., the Chinese version of the Social Support Rating Scale, Zarit Caregiver Burden Interview, Center for Epidemiologic Studies Depression Scale) should be described more succinctly.
Furthermore, within the first lines of the Discussion section, the authors do not need to repeat again what are the most relevant aims of this paper or what previous studies have demonstrated with regard to the main topic, as these contents have been already discussed extensively elsewhere. Here, i suggest to immediately focus on the most relevant findings of the study and their implications for the general readership.
Importantly, the most relevant limitations/shortcomings of the present study should be more directly and extensively addressed by the authors within the present study as the description of the main caveats is really poor and limited in the present version of the paper as currently presented.
Finally, the manuscript needs to be seriously reviewed by a native English speaker for the quality of language.
Author Response
Point 1: First, when the authors, within the Introduction section, reported that as a common mental disorder, depression may be long-lasting or recurrent, thus substantially impairing the individual’s ability to function in their daily life, they could also mention that depression is commonly associated with suicidal behavior and medical illnesses related to a further ehnanced suicide risk. In particular, medical disorders have been identified as significant risk factor for both suicidal behavior and suicidal ideation, especially among depressed patients. Specifically, the existence of previous mood disorder, prior and current history of medical disorders, and cognitive impairment were reported to be the most important risk factors for suicide. Thus, in order to briefly address this issue (although i understand that the link between medical disorders, depression, and suicidal behavior is not the main topic of this paper), i suggest to cite and rapidly discuss the systematic review paper published on Drugs Aging in 2015 (PMID: 25491561). Response 1: Thank you for your kind advice. We cite and rapidly discuss the systematic review paper published on Drugs Aging in 2015 (PMID: 25491561) to briefly address the link between medical disorders, depression, and suicidal behavior according to your suggestion (page 2, line 55-57). Point 2: Moreover, the most relevant psychometric instruments used in the present study (e.g., the Chinese version of the Social Support Rating Scale, Zarit Caregiver Burden Interview, Center for Epidemiologic Studies Depression Scale) should be described more succinctly. Response 2: Thank you for your kind advice. We have described the relevant psychometric instruments used in the present study (e.g., the Chinese version of the Social Support Rating Scale, Zarit Caregiver Burden Interview) more succinctly (page 3, line 126-130; page 4, line 136-141). Point 3: Furthermore, within the first lines of the Discussion section, the authors do not need to repeat again what are the most relevant aims of this paper or what previous studies have demonstrated with regard to the main topic, as these contents have been already discussed extensively elsewhere. Here, i suggest to immediately focus on the most relevant findings of the study and their implications for the general readership. Response 3: Thank you for your kind advice. We have deleted the first paragraph of the Discussion section according to your suggestion. In the new written beginning of Discussion section, we immediately focus on the most relevant findings of the study and their implications for the general readership (page 7, line229-232; page 8, 233-240). Point 4: Importantly, the most relevant limitations/shortcomings of the present study should be more directly and extensively addressed by the authors within the present study as the description of the main caveats is really poor and limited in the present version of the paper as currently presented. Response 4: Thank you for your kind advice. We rewrote the last paragraph of the Discussion part of our manuscript. So we have addressed the most relevant limitations or shortcomings of the present study more directly and extensively (page 9, line 290-297). Point 5: Finally, the manuscript needs to be seriously reviewed by a native English speaker for the quality of language. Response 5: Thank you for your kind advice. We proofread the manuscript in detail and improved the quality of language as possible as we can.
Round 2
Reviewer 1 Report
I am grateful for the opportunity to review this important research.
I made comments and edits directly in the PDF document of V2. See attached
Here is a gentle educational reminder about hypotheses:
"Hypothesis basics
A hypothesis is a suggested solution for an unexplained occurrence that does not fit into current accepted scientific theory. The basic idea of a hypothesis is that there is no pre-determined outcome. For a hypothesis to be termed a scientific hypothesis, it has to be something that can be supported or refuted through carefully crafted experimentation or observation. This is called falsifiability and testability, an idea that was advanced in the mid-20th century a British philosopher named Karl Popper, according to the Encyclopedia Britannica.
A key function in this step in the scientific method is deriving predictions from the hypotheses about the results of future experiments, and then performing those experiments to see whether they support the predictions." https://www.livescience.com/21490-what-is-a-scientific-hypothesis-definition-of-hypothesis.html
"So what are the words that we need to keep in mind? The hardest part about understanding scientific theories and hypotheses seems to be this: a hypothesis is never proven correct, nor is a theory ever proven to be true. Words like prove, correct, and true should be removed from our vocabulary completely and immediately." https://www.nsta.org/publications/news/story.aspx?id=52402

Author Response
Response to reviewer 1 comments
Point 1: I am grateful for the opportunity to review this important research. I made comments and edits directly in the PDF document of V2. See attached.
Response 1: Thank you for your kind advice, we have learned a lot from your advice. As we proofread the manuscript in detail, we checked and corrected all of the comments and edits in the PDF document of V2 accordingly as possible as we can (see details in the revised version of manuscript).
Point 2: Here is a gentle educational reminder about hypotheses:
“Hypothesis basics
A hypothesis is a suggested solution for an unexplained occurrence that does not fit into current accepted scientific theory. The basic idea of a hypothesis is that there is no pre-determined outcome. For a hypothesis to be termed a scientific hypothesis, it has to be something that can be supported or refuted through carefully crafted experimentation or observation. This is called falsifiability and testability, an idea that was advanced in the mid-20th century a British philosopher named Karl Popper, according to the Encyclopedia Britannica.
A key function in this step in the scientific method is deriving predictions from the hypotheses about the results of future experiments, and then performing those experiments to see whether they support the predictions." https://www.livescience.com/21490-what-is-a-scientific-hypothesis-definition-of-hypothesis.html
"So what are the words that we need to keep in mind? The hardest part about understanding scientific theories and hypotheses seems to be this: a hypothesis is never proven correct, nor is a theory ever proven to be true. Words like prove, correct, and true should be removed from our vocabulary completely and immediately." https://www.nsta.org/publications/news/story.aspx?id=52402
Response 2: Thank you for your gentle educational reminder about hypotheses, we have learned a lot from your advice. We have checked and removed words like prove, correct and true from our manuscript accordingly.

Reviewer 3 Report
In the revised paper, the authors addressed, in my opinion, most of the major questions raised by Reviewers improving both the main structure and quality of this manuscript. I have no further additional comments.
Author Response
Response to reviewer 3 comments
Point : In the revised paper, the authors addressed, in my opinion, most of the major questions raised by Reviewers improving both the main structure and quality of this manuscript. I have no further additional comments.
Response : Thank you very much.
